# Inoculation with *Rhizophagus irregularis* Does Not Alter Arbuscular Mycorrhizal Fungal Community Structure within the Roots of Corn, Wheat, and Soybean Crops

**DOI:** 10.3390/microorganisms8010083

**Published:** 2020-01-07

**Authors:** Sébastien Renaut, Rachid Daoud, Jacynthe Masse, Agathe Vialle, Mohamed Hijri

**Affiliations:** 1Institut de Recherche en Biologie Végétale, Département de Sciences Biologiques, Université de Montréal, 4101 Sherbrooke Est, Montréal, QC H1X 2B2, Canada; sebastien.renaut@gmail.com (S.R.); jacynthe.masse@umontreal.ca (J.M.); 2Microbial Biotechnology Applied to Agriculture and Environment, AgroBioSciences, Mohammed VI Polytechnic University, Lot 660, Hay Moulay Rachid, Ben Guerir 43150, Morocco; Rachid.Daoud@um6p.ma; 3Biopterre—Centre de Développement des Bioproduits, 1642 rue de la ferme, Sainte-Anne-de-la-Pocatière, QC G0R 1Z0, Canada; agathe.vialle@biopterre.com

**Keywords:** arbuscular mycorrhizal fungi, bioinoculants, crop production, field trials, community structure, amplicon sequencing

## Abstract

Little is known about establishment success of the arbuscular mycorrhizal fungal (AMF) inocula and their effects on a soil-indigenous community of AMF. In this study, we assessed the effect of introducing *Rhizophagus irregularis* DAOM-197198 in soil under field condition on the community composition of indigenous AMF in the roots of corn (*Zea mays*), soybean (*Glycine max*), and wheat (*Triticum aestivum*). Three field trials were conducted with inoculated and non-inoculated plots. Four to ten roots and their rhizosphere soil samples of two growth stages for corn and wheat, and one growing stage of soybean, were collected, totalling 122 root and soil samples. Root colonization was measured microscopically, and the fungal communities were determined by paired-end Illumina MiSeq amplicon sequencing using 18S rDNA marker. After quality trimming and merging of paired ends, 6.7 million sequences could be assigned to 414 different operational taxonomic units. These could be assigned to 68 virtual taxa (VT) using the AMF reference sequence database MaarjAM. The most abundant VT corresponded to *R. irregularis*. The inoculation treatment did not influence the presence of *R. irregularis*, or AMF community diversity in roots. This seems to indicate that inoculation with *R. irregularis* DAOM-197198 does not change the indigenous AMF community composition, probably because it is already present in high abundance naturally.

## 1. Introduction

Nutrient acquisition by plants is largely affected by organisms thriving in root and soil [1]. Among those essential soil microorganisms, arbuscular mycorrhizal fungi (AMF) in the phylum Glomeromycota are an ancient group of fungi that appeared together with the first land plants [2]. They form mutualistic associations with the roots of more than 80% of land plants including many important agricultural crops [3] and deliver mineral nutrients, especially phosphate to their host. In return, the fungi receive photosynthetically fixed carbon from the host plants. AMF are obligate biotrophs and they depend entirely on the carbon of their host plants. Their mycelia are formed by coenocytic hyphae. Inside the root, AMF mycelia form specialized exchange structures inside cortical cells called arbuscules. Some AMF species also form vesicles which are believed to play a role in nutrient storage, and sometimes spores in the intercellular space of the root cortex. In the soil, AMF mycelia extend their hyphae beyond the rhizosphere where they produce multinucleated spores, as well as auxiliary cells for some taxa [2]. Over the last 30 years, extensive research has been conducted to improve our understanding of how AMF deliver nutrients to their host. Ultimately, by harnessing this process and the factors that control nutrient cycling between the fungus and the plant, agricultural crop yield could be increased significantly while reducing the need to supply crops with system-external nutrients. This concept has tremendous potential to reduce the need for plant fossil and synthetic fertilizers which would help to mitigate the impact and cost of unsustainable nutrients mining [4,5,6]. As such, several beneficial AMF inoculants (microsymbionts) have been developed over the years hoping to enhance crop yield in a sustainable manner [7,8,9,10]. *Rhizophagus irregularis* DAOM 197198 (syn. *Glomus irregularis*, previously *Glomus intraradices*) is so far the most common AMF isolate that is produced commercially in axenic condition for two decades [10].

Despite extensive research, the findings are inconsistent concerning the establishment of introduced AMF inoculants in the field and eventual effectiveness in improving plant yield. Similarly, microbial inoculation has been documented to cause important changes in the number and composition of natural AMF communities present in the soil, but the impacts tend to be largely dependent on the inoculant and techniques used to quantify the AMF [11]. Arguably, a number of factors influence root colonization and plant benefits from AMF [12,13,14] making comparisons among studies difficult. Among these factors, competition between locally adapted indigenous AMF and introduced fungi is known to affect the establishment success of inoculants. In addition, indigenous AMF and their associated bacterial communities are often specific to the soil type, historical land use, and specific crop plants species present [15]. Moreover, individual plants and crop cultivars and genotypes can differ in their response to AMF inoculation [16]. For their part, Kokkoris et al. [14] showed that the establishment of AMF was site specific, and therefore neither cropping nor inoculation practices could predict inoculation success in the field. Finally, measuring AMF activity and effects on crop performance in the field is difficult and often not standardized among studies, leading to spurious outcomes which are difficult to reproduce [6].

Given these mixed results, Ryan and Graham [6] concluded that in the past, the literature has often presented an overly optimistic view of the importance of AMF in positively impacting yield. Conversely, studies frequently provide a pessimistic view of the impact of common agricultural practices on AMF communities and are not necessarily accurate. As such, Ryan and Graham [6] stated that scientific evidence that farmers should consider abundance and diversity of AMF when managing crops is scarce. From the biodiversity conservation point of view, Hart [9] further warned of the unintended consequences of introduced (and putatively invasive) inoculants in natural systems, since they can disrupt local ecosystem functioning. Therefore, Ryan and Graham [6] recommended that more critical research and new analytical approaches are needed before AMF inoculation becomes a cost-effective way to increase crop yield that is widely accepted within the scientific community and the agribusiness. In general, we need to know more about the multifarious roles played by AMF in agroecosystems. Concurrently, Rillig et al. [5], for example, argue among other points similar to those by Ryan and Graham [6] that AMF are not just important for yield, but also for system performance and sustainable soil use. In addition, several other factors such as inoculant quality, formulations, and mode of application need to be considered when assessing AMF inoculation. For example, Hijri [10] reported that no or negative effects on potato yield were mainly due to the wrong application of the inoculants by farmers. A study of the quality of 12 AMF inoculants by testing them on maize showed contrasting results in terms of AMF taxa composition, efficiency of root colonization, and yield promotion when the evaluation was in sterilized sand as opposed to natural soil [17].

Finally, several crops have been shown to respond positively (i.e., show increased yield) in a consistent manner to AMF inoculation [10,18]. Regardless of whether AMF inoculation becomes a widely accepted agricultural practice, the fact remains that the interactions among introduced and indigenous AMF, soil microbes, plant hosts, and environmental or agricultural conditions are multifaceted, complex, and still poorly understood [6].

So far, all known AMF are obligate symbionts, and therefore not culturable without a host plant. Methods for AMF identification rely mostly on the morphology of spores such as the mode of spore formation, size, shape, color, hyphal attachment, and reaction to the Melzer stain. These morphological analyses are time consuming and require highly specialized expertise due to the lack of morphological traits. Therefore, AMF taxon recognition relies largely on molecular approaches [19]. For AMF, amplification and sequencing of a divergent domain of the small subUnit of the ribosomal RNA (SSU rRNA sequence marker) that is short enough to be covered by high throughput sequencing and long enough to provide sufficient taxonomic resolution has become the standard for AMF community analyses [15,20]. A universal system of nomenclature has been developed that defines similarity groups based on this SSU rRNA (18S rRNA) sequence marker (MaarjAM database by Öpik et al. [21]), the so-called Glomeromycota virtual taxa (VT), in order to make datasets comparable among studies.

In this investigation, we aim to assess the influence of inoculating corn (*Zea mays*), wheat *(Triticum aestivum*), and soybean *(Glycine max*) with the AMF *Rhizophagus irregularis* DAOM-197198 on the community structure of indigenous AMF in the roots. Specifically, we tested if inoculation had an effect on the following: (1) The presence of indigenous *R. irregularis*, (2) glomeromycotan diversity (alpha diversity), and (3) VT turnover among sites (beta diversity).

## 2. Materials and Methods

### 2.1. Field Test Study Design

Corn (*Zea mays*), cultivar Elite 49A12 Cruiser Max Quattro, was sown in a field located at St-Elzéar, Québec, with a loamy sand soil on 9 May 2015, as part of a soybean and corn rotation. The field was subdivided into six plots and each plot contained 24 rows spaced by 76 cm. Three plots were inoculated with Myke Pro Liquid (Premier Tech, Rivière-du-Loup, QC, Canada) containing *R. irregularis* (syn. *Glomus irregulare*), isolate DAOM-197198 (1.9 L of liquid inoculant added to a 70 gallon tank, inoculation rate 325 spores/m^2^). Three plots were not inoculated and served as a control. All plots received the same fertilization and pest management treatments as used under conventional agricultural practice (details on fertilization and pest managements can be found in Hijri [10]). Two growth stages were sampled. The first sampling (V4 stage) was done on 10 June 2015 and included 21 randomly collected inoculated plants and their rhizosphere soil, and five non-inoculated (control) plants. The second sampling (V8 stage) was done on 25 June 2015, when 23 inoculated plants and five non-inoculated (control) plants were taken.

Wheat (*Triticum aestivum*) cultivar Touran was sown in a field located at Ste-Hélène de Kamouraska, Québec, with a loamy soil on 15 May 2015. A twenty-two meter row was inoculated with Myke Pro PS3 (*R. irregularis* isolate DAOM-197198, inoculation rate 325 spores/m^2^; Premier Tech). Two growing stages were also sampled. The first sampling was done on 15 June 2015 and included 16 inoculated and four non-inoculated (control) plants which were sampled randomly along with their root systems. The second sampling was done on 3 July 2015, when 17 inoculated and five non-inoculated (control) plants were taken randomly, along with their root systems.

Soybean (*Glycine max*) cultivar Pioneer 90Y01 was sown in an experimental field located at Notre-Dame-du-Mont-Carmel (Québec) with a loamy sandy soil on 6 May 2015, as part of a soybean and corn rotation. The field was subdivided in ten plots and each plot contained 12 rows spaced by 30 cm. Five plots were inoculated with Myke Pro Soybean Liquid (Premier Tech, 1.9 L of inoculant (*R. irregularis*, isolate DAOM-197198) in an 80 gallon tank, inoculation rate was 125 spores/m^2^). Five plots were not inoculated and served as a control. The sampling was done on 25 June 2015 when 20 inoculated and six non-inoculated (control) plants were randomly taken, along with their root systems. Plants which were sampled at the early growth stage given that mycorrhizal colonization was already assessed to be high (see next section on root staining and determining mycorrhizal colonization) in both inoculated and control plants.

Climatic type of the region where the three fields are located is temperate cold. The growing season typically lasts for approximately 5 months, from May to October. The climatic conditions are the following: the average daily high temperature during the warm season (3.7 months, from May to September) above 18 °C. The hottest month of the year is July, with an average high of 24 °C and low of 15 °C. The average daily high temperature during the cold season (3.3 months, from December to March) is below −1 °C. The coldest month of the year is January, with an average low of −15 °C and high of −7 °C. The rainy period of the year lasts for 9.6 months, from March to December, with a sliding 31 day rainfall of at least 13 mm. The most rain falls during the 31 days centered around June 29, with an average total accumulation of 84 mm.

None of the three fields (corn, wheat, and soybean) had been previously inoculated with *R. irregularis* (isolate DAOM-197198). In total, 122 different samples were processed under the current experimental design shown in Table 1.

### 2.2. Staining and Determination of Fungal Root Mycorrhizal Colonization

Roots were rinsed under tap water to remove soil particles and stained using the ink and vinegar method described by Vierheilig et al. [22]. Root colonization was assessed semi quantitively under a Zeiss binocular (Oberkochen, Germany) on a scale from 0 (no root colonization) to 3 (very strong colonization).

### 2.3. DNA Extraction from Soil and Illumina MiSeq Sequencing

Prior to DNA extraction, 250 mg of soil sample were suspended in 400 mL of 0.75% hydrogen peroxide and manually agitated until complete disintegration of the aggregates. The suspension was then wet sieved under vibration and water pressure (Retsch AS200 control, Haan, Germany) with the following parameters: 4 successive sieves (2.00 mm, 1.00 mm, 45 µm, and 38 µm) and increasing vibrations of 0.25 up to 1.87 Hz every 40 s for 5 min. The collected soil fraction retrieved in the 38 µm sieve was transferred to a 20 µm polyester tissue to remove excess water. Finally, 100 mg of this filtrated sieved soil was transferred to microtubes for DNA extraction. The total gDNA was extracted using a NucleoSpin^®^ Soil isolation kit with some modifications (Mascherey-Nagel, Düren, Germany) as described by Badri et al. [23]. The total DNA of sieved soil samples was extracted in triplicate.

The DNA of arbuscular mycorhizal fungi (AMF) was amplified using 18S rRNA gene primers AML1 (5′-ATC AAC TTT CGA TGG TAG GAT AGA-3′) and AML2 (5′-GAA CCC AAA CAC TTT GGT TTC C-3′) using polymerase chain reaction (PCR), which generates amplicons of ~800 base pairs (bp) in length. Nested PCRs were then performed in order to ensure that no off-target amplicons were sequenced using an in-house set of internal primers with CS adapters (shown in bold): nu-SSU-0595 (5′-**ACACTGACGACATGGTTCTACA**CGGTAATTCCAGCTCCAATAG-3’ and nu-SSU-0948 (5′-T**ACGGTAGCAGAGACTTGGTCT**TTGATTAATGAAAACATCCTTGGC-3′) yielding an amplicon of ~400 bp. DNA samples were then barcoded, pooled, and pair-end sequenced (2 × 300 bp) using an Illumina MiSeq sequencer (San Diego, CA, USA) at the Genome Québec Innovation Centre, McGill University, Montreal, QC, Canada.

### 2.4. Bioinformatics

Sequences were demultiplexed by the sequencing facility (Genome Québec Innovation Centre). Following this, we developed an in-house pipeline to generate a table of operational taxonomic units (OTUs) based on the sequence identity and clustering method with a conservative threshold, as described below. Note that scripts along with specific parameters used in the analyses are available at (https://github.com/seb951/Premier_Tech_inoculation). Briefly, sequences were trimmed based on quality and length to eliminate potential Illumina adaptor contamination, using TRIMMOMATIC-0.36 [24]. Then, the paired-end reads were merged and trimmed again in MOTHUR [25]. A second trimming step, after merging, was necessary in order to remove homopolymers (length > 10) and sequences with ambiguous bases. Following this, chimeras were detected and removed with VSEARCH [26]. In QUIIME [27], taxonomy was assigned and reads not assigned to Glomeromyceta were filtered out, afterwards an OTU count table was generated. Finally, samples that contained less than five hundred OTU-assigned sequences were omitted from the analysis, which were run on the OTU abundance after rarefication using MUTHER pipeline.

In order to reduce the complexity of the dataset, OTUs were further grouped into virtual taxa (VT) using BLASTN [28] and the MaarjAM database (34,261 sequences in the class Glomeromycetes clustered into 704 virtual taxa, downloaded on 15 June 2018). Here, a virtual taxon is defined as a phylocluster of similar small subunit ribosomal RNA (SSU rRNA) gene sequences at 97% sequence. All VT are anchored to a type sequence, with its VT identity preserved in time, which guarantees standardization and taxonomic assignment, and thus comparability among studies [21]. For VT assignment, several OTUs had to be merged into the same VT. We also verified that the inoculant (*R. irregularis* DAOM-197198) matches best to VTX00113 and VTX00114 using BLASTN searches (*e*-value of 0 and >99% homology against the *R. irregularis* DAOM-197198 genome). However, given the short length of the amplicons, the ubiquitous nature of AMF in the environment, and their very low endemism [29], the current experimental design cannot distinguish between indigenous *R. irregularis* and the inoculant itself (DAOM-197198). We amplified the 18S rDNA fragment from pure DNA of the *R. irregularis* (DAOM-197198) using the same primers nu-SSU-0595 and nu-SSU-0948 that were used for amplicon sequencing. The PCR product was then sequenced using the Sanger method. Sequences of DAOM-197198 matched VTX00113 and VTX00114 with only one nucleotide difference.

### 2.5. Statistical Analyses

All statistical analyses were performed in R 3.4.4 version [30] and detailed scripts of analysis are available at (https://github.com/seb951/Premier_Tech_inoculation). First, we tested if the inoculation treatment had a significant effect on the presence of *R. irregularis* (virtual taxa VTX00113 and VTX00114). We used a linear mixed-effect (LMM) model in the R package NLME [31], which is more appropriate than an ANOVA to account for the unbalanced block design. The relative abundance of VTX00113 and VTX00114 were square root transformed in order to satisfy the assumption of normality of the residuals in the LMM statistical framework. The crop species corn, wheat, and soybean growth stage interactions were tested using a linear model (LM), given that the current study design did not allow testing for the interaction directly in the LMM framework.

Community analyses were used to check the effect of inoculation, crop species, and growing stage on the relative abundance of all virtual taxa. VT in each of the crop species and inoculation treatments (inoculated and non-inoculated) were used for visualization. To reorganize more general patterns, we also visualized the relative abundance at the genus, family, and order level, using MaarjAM taxonomy [21]. AMF diversity (α-diversity) was calculated at each site using the inverse Simpson diversity index in the R package VEGAN [32]. The effect of crop species, growth stage, and treatment were assessed using a linear mixed-effect (LMM) model in the R package nlme [31] to account the unbalanced block design. The value of VT alpha diversity was log transformed in order to satisfy the assumption of normality of the residuals of the LMM. The crop species corn, wheat, and soybean growth stage interactions were tested using a linear model (LM), given that the current study design did not allow testing the interaction directly in a LMM framework.

To analyze the effect of inoculation, crop species, and growth stage on the community structure (beta diversity) and composition, we conducted a permutational multivariate analysis of variance (PERMANOVA, 10,000 permutations) on the Hellinger-transformed count data in VEGAN [32]. Crop species, growth stage, treatment, and block were factored into the model. We also added a crop species * growth stage interaction in the model after testing that other interactions in the model were nonsignificant. Postdoc tests using the R package RVAideMemoire, were conducted [33] to test the effect of the growing stage (early vs. late) and crop species on the community structure and composition. We performed a principal coordinate analysis (PCoA, R package APE, [34]) using the Bray–Curtis dissimilarity matrix of Hellinger-transformed community data to check whether the AMF communities clustered according to crop species, growth stage, or the inoculation treatment.

Redundancy analysis (RDA) using Hellinger-transformed VT data was carried out in VEGAN [32] to identify the factors (species, growth stage, and treatment) that determined the AMF communities in the rhizosphere soil of the three crop plant species. We tested the significance of the model, the axes, and the environmental variables (crop species * growth stage * treatment) using a permutation test (1000 permutations). AMF community differentiation according to the significant factors, crop plant species and growth stage, was visualized, using 95% confidence ellipses around the centroids of the communities of each factor level.

## 3. Results

### 3.1. Mycorrhizal Root Colonization

In corn, mycorrhizal root colonization was low at the first sampling and high in the second sampling stage, whereas, in wheat, colonization was low in the first stage and moderate in the second stage (Table 1). Mycorrhizal root colonization was high in both inoculated and control soybean plants. No significant difference on mycorrhizal root colonization was observed between inoculated and non-inoculated plots (*p*-value > 0.05) (Appendix A, Appendix A).

### 3.2. AMF Diversity

A total of 8.18 million paired-end raw reads were obtained for all 122 samples (33,549 ± 9298 reads per sample, 95% CI) and after quality trimming of the raw reads, 8.00 million paired-end reads (32,786 ± 9097 reads per sample, 95% CI) were kept and further processed. Paired-end reads were merged, trimmed, and filtered to remove reads with homopolymers and ambiguous sites, resulting in 6,711,213 merged reads (mean length of 395 ± 24 bp, 95% CI), which corresponded well with the predicted amplicon size of 400 bp. These were grouped into 414 unique OTUs at 96% sequence identity after chimeras were removed and further clustered into 68 virtual taxa (VT) based on the MaarjAM database [21] (Appendix A, Appendix A). Forty-two (62%) of these VT were assigned to *Glomus* (now *Rhizophagus*) taxa. Seven samples (E10_BE, E3_BE, E3_BT, E4_BE, E5_BE, E6_BE, and T3BT_BT) were dropped from the statistical analyses because they yielded too few sequences.

The most abundant taxa VT (VTX00113) corresponded to *R. irregularis* based on blastn and represented 38% of all reads (Appendix A). Inoculation did not affect the abundance of VTX00113 (LMM, *p*-value = 0.63), while crop species had a large effect (LMM, *p*-value < 0.0001), with corn showing the greatest relative abundance, followed by wheat (Figure 1A). Growth stage had a marginal significant effect (LMM, *p*-value = 0.046), with late stage having a greater relative abundance (Figure 1 and Appendix A, Appendix A). The interaction between growth stage and crop species was marginally nonsignificant (LM, *p*-value = 0.053). The other VT (VTX00114) assigned to *R. irregularis* represented 0.2% of all reads and showed similar patterns to VTX00113 (LMM, crop species effect *p*-value < 0.003 and treatment effect *p*-value < 0.19, Figure 1B).

Comparing the relative abundance of VT among the three crop species and two inoculation treatments (Figure 2) showed that taxa of the genus *Rhizophagus* and *Funneliformis* (VTX00113, VTX00067, VTX00093, and VTX00115) dominated the communities, but they also comprised other *genera* such as *Acaulospora, Diversispora*, *Claroideoglomus,* and other genera as well (Figure 3).

The AMF alpha diversity (species diversity at local scale in a single ecosystem or specific area) differed among the crop species (LMM, *p*-value = 0.0033) alone; soybean having the highest alpha diversity (Figure 4). Inoculation and growth stage did not have an impact on AMF alpha diversity (LMM, *p*-value = 0.93 and *p*-value = 0.12, respectively). The interaction between crop species and growth stage was also found to be nonsignificant (LM, *p*-value = 0.23).

### 3.3. AMF Community Structure

On the basis of the PERMANOVA statistical framework, crop species (*p*-value < 0.0001 and nperm = 9999), growth stage (*p*-value = 0.001and nperm = 9999) and their interaction (*p*-value = 0.017 and nperm = 9999) had a significant effect on AMF community structure (beta diversity) between the two growing stages. However, here also the inoculation had no effect on the community structure (*p*-value = 0.85 and nperm = 9999). The effect of crop species can also be seen visually through a principal coordinate analysis, as depicted in Figure 5. In addition, based on a permutation post hoc MANOVA. The effect of growth stage was significant in the wheat crop (*p*-value = 0.0001), marginally in corn (*p*-value = 0.07), and in soybean only one growth stage was sampled.

RDA analysis was conducted with all three environmental variables (crop species * growth stage * treatment). Given that inoculation did not show a significant effect (*p*-value = 0.3), we conducted the RDA without this factor removing the term from the model. Both the model (*p*-value = 0.001), the first three axes (*p*-value = 0.001, *p*-value = 0.001, and *p*-value = 0.006, respectively), the crop species (*p*-value = 0.001), and the growth stage (*p*-value = 0.006) were significant. We identified several taxa that seem characteristic for one of the three crop species and for growth stage (Figure 6), such as VTX00113 (*R. irregularis*) and VTX00062 (*Diversispora sp.*, Figure 2) that were characteristic for early corn growth stage or VTX00115 (*Glomus* sp. synonym *Rhizophagus* sp. MO−G13, in Figure 2) and VTX00067 (*Glomus mosseaea*, synonym *Funneliformis mosseaea*) which were abundant under wheat cropping.

## 4. Discussion

In this study, we assessed under field conditions, the influence of inoculated *Rhizophagus irregularis* DAOM-197198 on indigenous AMF communities using high throughput sequencing of 18S rRNA genes. Inoculation of field crops with beneficial microbes attracts considerable attention [4,5,6]. However, the impacts of inoculation tend to be specific to the application of the inoculants, the techniques used to describe the AMF communities and the crop species as reviewed in Trabelsi et al. [11]. Here, we found no evidence that *R. irregularis* DAOM-197198 inoculation influenced the relative abundance of indigenous *R. irregularis*, AMF alpha diversity, or AMF community composition and structure. It is likely that this species was already naturally present in significant amounts in these fields.

While AMF inoculation can have a positive impact on plants growth in inoculum-limited soil such as degraded soils, this cannot necessarily be extrapolated to inoculation performance in most agricultural soils that have well-established resident AMF communities [9]. Hart et al., 2017 [9] suggested that inoculants such as *R. irregularis* can have unintended consequences and pose a threat to soil and plant biodiversity. However, in the current study, inoculation did not appear to impact the indigenous AMF, likely because the AMF community has co-evolved and is already adapted to indigenous *R. irregularis* present in the soil. Nevertheless, inoculation could still have affected other non-AMF microbial taxa in the rhizosphere that we did not assess with the current experimental design.

The growth stage (early vs. late in maize and wheat) did not affect the Simpson diversity of VT, but significantly affected community structure (beta diversity). This is corroborated by the microscopically determined root colonization intensity that did not differ depending on the growth stage (little colonization in first stage of corn and wheat, see Table 1). In fact, as root colonization increased (inspected microscopically, and verified by community sequencing and the relative abundance of VTX00113 and VTX00114 (Figure 1A,B) overall alpha diversity relatively decreased (Figure 4) indicating that specific AMF such as *R. irregularis* outcompete other AMF and at least reduce the AMF diversity.

AMF community structure was more dependent on the crop plant species (corn, soybean, and maize) than that of the inoculation treatment. Indigenous AMF and their associated bacterial communities are often specific to the soil type, historical land use, and crop species present [15]. Plant roots are known to actively select their microbiome from the surrounding environment [35] resulting in site- and condition-dependent holobionts, composed of the plant and its microbiome. While plants certainly have an influence on the current communities, given that plants were grown in different locations, we cannot disentangle the plant species effect from the soil effect.

Admittedly, one caveat of our current sequencing strategy is that it did not allow distinguishing between indigenous *R. irregularis* and the inoculant itself (*R. irregularis* DAOM-197198). This strain was first isolated from indigenous AMF in Southern Québec (Pont Rouge, QC, Canada), where the current experimental plots were situated, before being selectively bred as a commercial inoculant. Consequently, the inoculant and indigenous *R. irregularis* are likely to be phylogenetically related, and thus indistinguishable using current high throughput MiSeq sequencing on the SSU rRNA marker. This sequencing approach yields amplicons of relatively small size (~400 bp), which do not have discriminating power between closely related genotypes with only one base pair difference. In addition, OTUs were grouped into VT using the MaarjAM database [21]. The MaarjAM database, which is increasingly adopted as a universal system of nomenclature for environmental sequence identification of Glomeromycota SSU rRNA gene sequences, allows comparisons and standardization among studies. However, much like any similarity-based clustering-based method, the VT approach reduces the complexity of the dataset, and consequently fine grain information on specific genotypic differences is lost.

While AMF are known to be globally distributed and to show very limited endemism even at a continental scale [29], analyses of coding genes indicate that each AMF isolate often possesses a unique genome, a telltale sign of local adaptation [36]. While this study showed that inoculation did not have an impact on indigenous AMF communities, we have merely started to understand the biogeography and evolution of glomeromycetes both at a local and global scale [9].

## Figures and Tables

**Figure 1 microorganisms-08-00083-f001:**
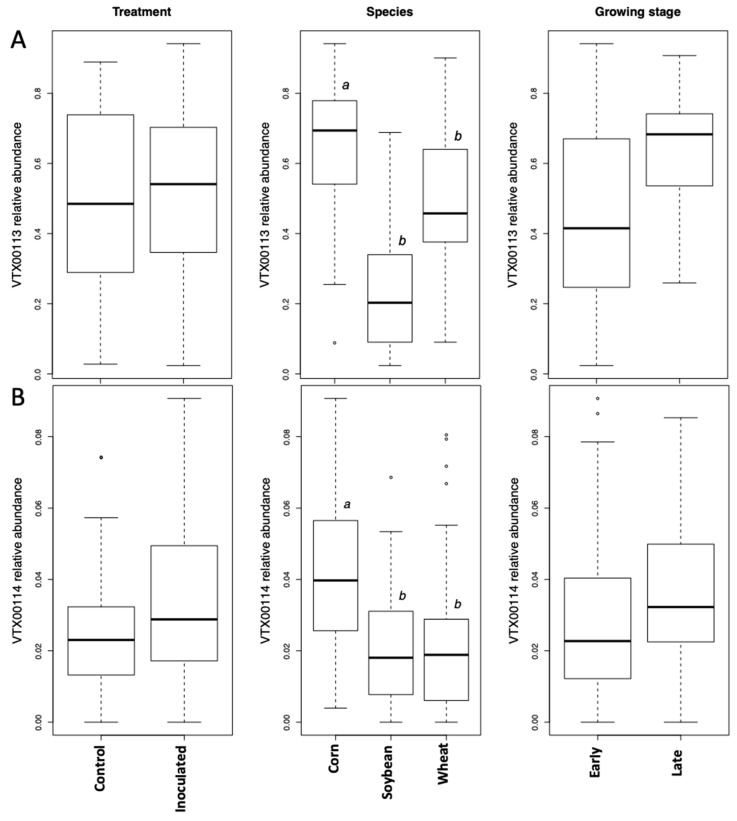
Relative abundance of the most abundant taxon *Rhizophagus irregularis* (VTX00113 (**A**) and VTX00114 (**B**). *R. irregularis* was represented by an average of 38% of reads across all samples in all three species (wheat, corn, and soybean) tested. Crop species and growth stage had a significant effect on the presence of VTX00113 and VTX00114. Letters a and b show statistically significant differences between variables.

**Figure 2 microorganisms-08-00083-f002:**
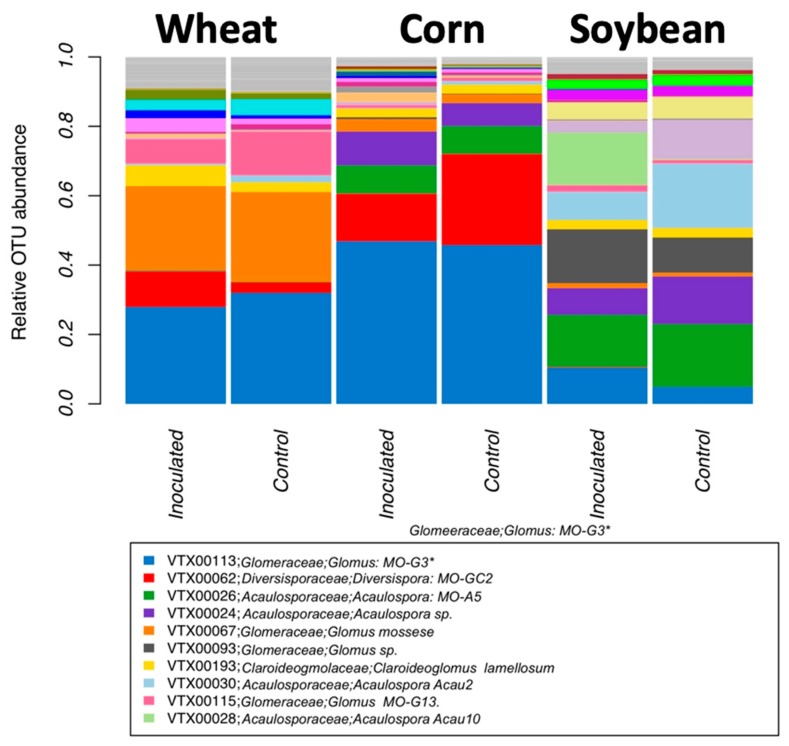
Mean relative abundance of all virtual taxa (VT) and taxonomic affiliation according to the MaarjAM [20] database for the different inoculation treatment and crop species.

**Figure 3 microorganisms-08-00083-f003:**
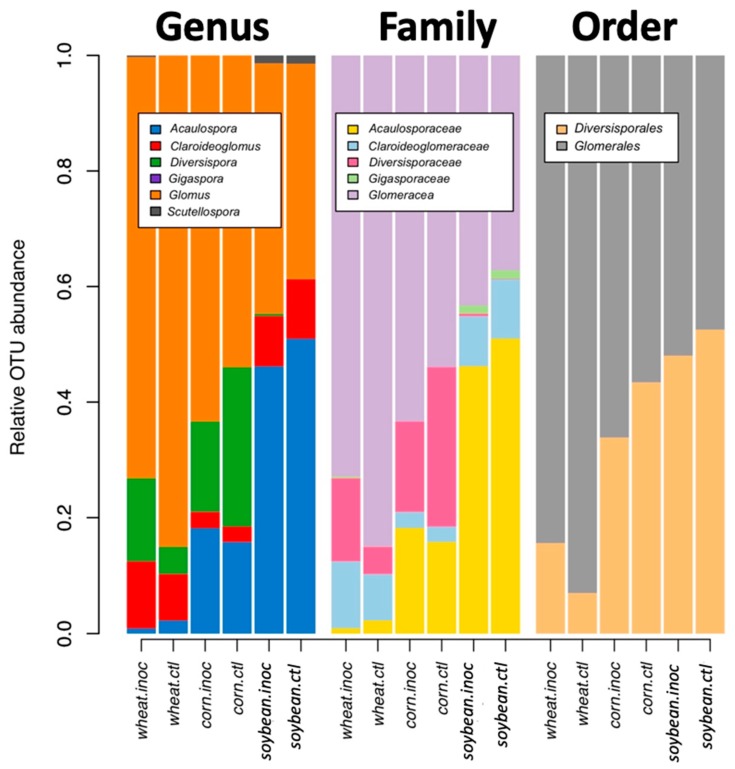
Mean relative abundance of the virtual taxa (VT) grouped by genera, order, and family according to the MaarjAM [20] database for the different inoculation treatment and crop species.

**Figure 4 microorganisms-08-00083-f004:**
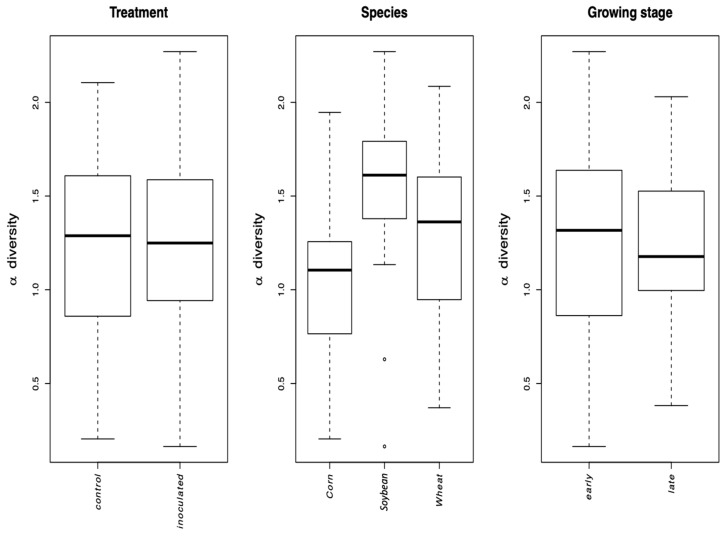
Simpson diversity of virtual taxa (VT) of the different inoculation treatments, crop plant, species, and growth stages. Crop species and growth stage had a significant effect on diversity. Median values and interquartile ranges are shown in the plots.

**Figure 5 microorganisms-08-00083-f005:**
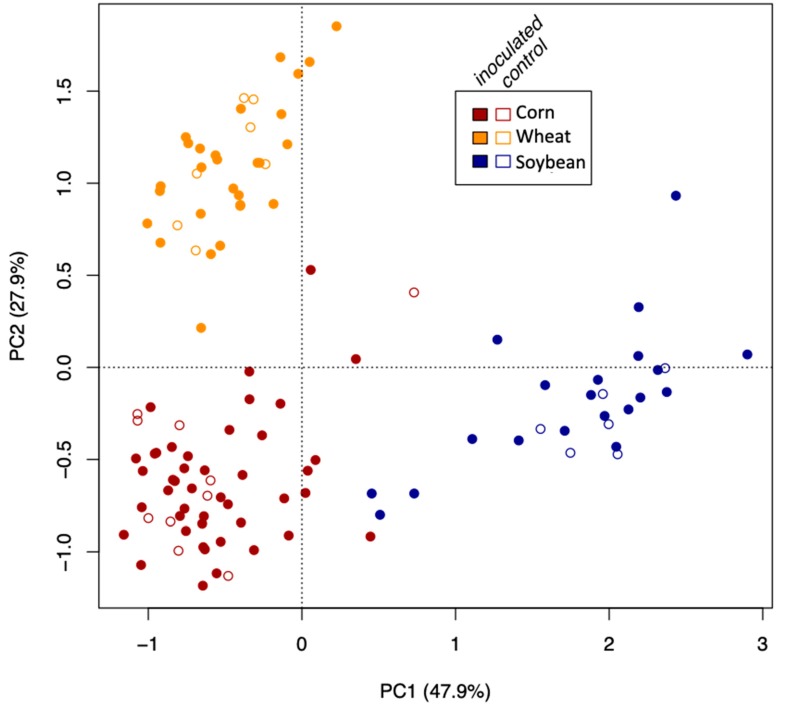
The effect of crop species on the AMF community structure based on principle coordinate analysis (PCoA) of the square root transformed for all the soil samples. Color coding is used to distinguish the three crop plant species (wheat, corn, and soybean) and open and filled circles for the inoculated and non-inoculated treatments.

**Figure 6 microorganisms-08-00083-f006:**
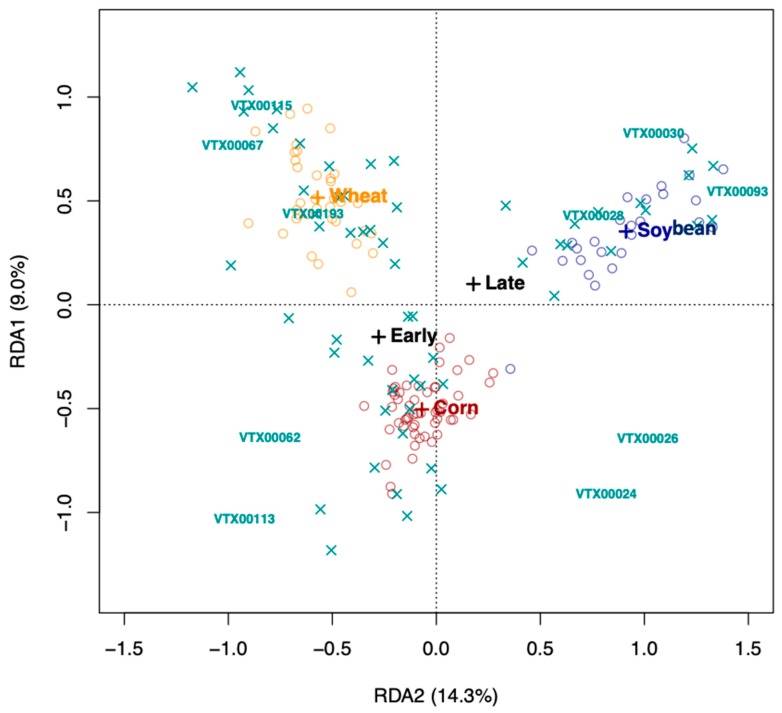
Redundancy analysis (RDA). Note that only the ten most abundant virtual taxa (VT, “X” signs) are labeled, “O” represents samples, and “+” represents centroids of the crop species (wheat, corn, and soybean) and growth stage (early vs. late). The inoculation treatments are not highlighted, as samples did not group accordingly (see Figure 5). The percentage of variance explained is indicated in brackets after the axis labels.

**Table 1 microorganisms-08-00083-t001:** Sampling design and root colonization *.

Crop	Corn	Corn	Soybean	Wheat	Wheat	Total No. Samples
Growing Stage	Early Stage (4 Weeks)	Late Stage (8 Weeks)	Early (4–5 Weeks)	Early (25 Days)	Late (39–40 Days)	
No. of blocks	3	3	5	1	1	
No. of samples/blocks	7	7 or 8	4	16	17	
No. of controls (non-inoculated plants)	5	5	6	4	5	
Total No. of samples	26	28	26	20	22	122
Root colonization						
inoculated	0–2	2–3	2–3	0–1	2–3	
control	0–1	2–3	2–3	0–1	2–3	

* root colonization was assessed semi-quantitatively for each sample on a scale of 0 (no colonization), 1 (little), 2 (moderate), and 3 (heavy colonization).

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
