# Peer review of "Inoculation with Rhizophagus irregularis Does Not Alter Arbuscular Mycorrhizal Fungal Community Structure within the Roots of Corn, Wheat, and Soybean Crops"

_microorganisms, 2020, doi:10.3390/microorganisms8010083_

Round 1

Reviewer 1 Report

The authors have presented original research, the manuscript is of interest to the mycorrhizal community and the research design is mostly acceptable.  The conclusions are careful and generally supported by the results.

The paper requires however some important clarifications and additions, particularly in the methodology part.

A. General comments

Basic climatic conditions must be provided for the three sites (what type of general climate: temperate, humid, continental, maritime, etc). Precipitation data (mm/year etc) are crucial for any crop related paper. Precipitation and climate data are easily obtainable from national and provincial repositories. On Line 111 the authors state "All plots received the same fertilization and pest management treatments as used under conventional agricultural practice".
Both pest management and fertilization are important factors for mycorrhizal establishment and even if they are not in the remit of the current research study (as they are constant), they must be specified, just like soil conditions, climate, etc. Keep in mind that "conventional agricultural practice " may have a very different meaning for a diverse international readership. Line 113: ..."and their rhizosphere soil"...: Can this be specified by volume or mass? When describing the research sites, it is good practice to include the size of the fields and plots (square meters, hectares, etc) It is not clear why the inoculation rate was different for Site 3 (soybeans) which makes the comparison with the other two sites more difficult p9: Order and family in the column title of Fig 3 must be swapped. p9: For consistency, use singular, also in the figure legend: genus instead of genera Line 291: "95% confidence ellipses are drawn around the centroids of the crop plant species." Missing???

B Specific comments

Line 16: introducing the Rhizophagus Line 34-35: soil [1],. Among ... Line 53: competition between with locally Line 61: spurious outcomes which are difficult ... Line 65: AMF communities which may not necessarily true be accurate. Line 67-68: warned of the fact that the unintended consequences ... Line 78: evaluated the quality 82: Finally, several 84: a widely an accepted 88: Methods 90-91: because of due to the lack of morphological AMF possess limited traits. 96: the so-called 97: a system making in order to make datasets ... 120: plants which were 135: is in 150: protocol as 180: guarantees 181: comparability among studies 183: matches 187: amplified the 189: the Sanger method 230: In corn, mycorrhizal colonization root colonization was low 231: while in the wheat 277: Crop species identity and growth stage had a significant  289: Fill gaps in this sentence: "Principle Coordinate Analysis (PCoA) of the square-root transformed of the all soil samples." 296: show a significant 310: accordingly 315: genes 317: to descried describe the AMF communities and to the crop species inoculated as reviewed in ... 318: that the 319: delete the [twice] 320: it this species 324: such as R. irregularis may have unintended consequences and does pose a threat 340: holobionts 341-342: plants were growth grown in different locations and different soils as well, we cannot ... 346: selectively 356: is lost  

Author Response

We are very grateful to the reviewer for taking the time to think about this work seriously and provide us with the opportunity to present a more concise paper including all the reviewer comments. We think that this revision strengthen the manuscript and we feel that these suggestions have greatly improved the manuscript

We have given the specific details below of how the manuscript has been changed or how we have addressed the reviewer’s comments.

General comments

Basic climatic conditions must be provided for the three sites (what type of general climate: temperate, humid, continental, maritime, etc). Precipitation data (mm/year etc) are crucial for any crop related paper. Precipitation and climate data are easily obtainable from national and provincial repositories.

A paragraph summarizing climatic condition was added.

On Line 111 the authors state "All plots received the same fertilization and pest management treatments as used under conventional agricultural practice".
Both pest management and fertilization are important factors for mycorrhizal establishment and even if they are not in the remit of the current research study (as they are constant), they must be specified, just like soil conditions, climate, etc. Keep in mind that "conventional agricultural practice " may have a very different meaning for a diverse international readership.

A reference was added to provide information on fertilization and pest management.

Line 113: ..."and their rhizosphere soil"...: Can this be specified by volume or mass? When describing the research sites, it is good practice to include the size of the fields and plots (square meters, hectares, etc).

The information was added.

It is not clear why the inoculation rate was different for Site 3 (soybeans) which makes the comparison with the other two sites more difficult.

Formulation of commercial AMF inoculants is done for each crop. This explains why the number of propagules was different between crops.

p9: Order and family in the column title of Fig 3 must be swapped.

Changed.

p9: For consistency, use singular, also in the figure legend: genus instead of genera

Changed.

Line 291: "95% confidence ellipses are drawn around the centroids of the crop plant species." Missing???

This sentence was omitted from the legend.

B Specific comments

Line 16: introducing the Rhizophagus

Done

Line 34-35: soil [1],. Among ...

Done

Line 53: competition between with locally

Done

Line 61: spurious outcomes which are difficult ...

Done

Line 65: AMF communities which may not necessarily true be accurate.

Done

Line 67-68: warned of the fact that the unintended consequences ...

Done

Line 78: evaluated the quality

Done

82: Finally, several

Done

84: a widely an accepted

Done

88: Methods

90-91: because of due to the lack of morphological AMF possess limited traits.

Done

96: the so-called

Done

97: a system making in order to make datasets ...

Done

120: plants which were

Done

135: is in

Done

150: protocol as

Done

180: guarantees

Done

181: comparability among studies

Done

183: matches

Done

187: amplified the

Done

189: the Sanger method

Done

230: In corn, mycorrhizal colonization root colonization was low

Done

231: while in the wheat

Done

277: Crop species identity and growth stage had a significant 

Done

289: Fill gaps in this sentence: "Principle Coordinate Analysis (PCoA) of the square-root transformed of the all soil samples."

Done

296: show a significant

Done

310: accordingly

Done

315: genes

Done

317: to descried describe the AMF communities and to the crop species inoculated as reviewed in ...

Done

318: that the

Done

319: delete the [twice]

Deleted

320: it this species

Done

324: such as R. irregularis may have unintended consequences and does pose a threat

Done

340: holobionts

Done

341-342: plants were growth grown in different locations and different soils as well, we cannot…

Done

 346: selectively

Done

356: is lost

Done 

Reviewer 2 Report

Brief summary:

In this manuscript, authors conducted field trials with three different crops (corn, wheat and soybean) inoculated or not with Rhizophagus (=Glomus) irregularis. They collected several samples from roots and rhizospheres of each crop and condition and determined fungal communities by sequencing 18S rDNA marker. After an extensive statistical analysis, authors determined that inoculation with R. irregularis does not change the native arbuscular mycorrhizal fungi community composition. I think that English style is good (I´m not an English native speaker). However, some parts of the manuscript could be improved, since some statistical analyses and data are not displayed, and some sections of the paper should be edited and/or extended (see below). In my opinion, data displayed in this paper are interesting and useful, but their relevance is partially obscured by several points that should be solved. For this reason, I suggest reconsideration after major revision.

Major points:

Some statistical analyses and data are not showed in the manuscript. Inclusion of this new data will complete the overall vision of the conclusions extracted from experiments. Specifically:

In Figure 1: include a second panel with relative abundance of taxon VTX00113.

Repeat same analysis showed in figures 2 and 3 for the different growing stages of wheat and corn crops.

Figure 5: include a second panel with PCoA analysis for growing stages of wheat and corn crops.

Line 242-246: provide a supplementary file with 414 unique OTUs and 68 VTs identified grouped by crops, growing stages and treatments.

Some sections of the paper should be revised:

Lines 62-81: This paragraph fits better in the Discussion section.

In the Introduction section information about the microsymbiont is missing.

Result section is too short and could be sorted in several subsections.

Minor points:

Unify crop names: soybean vs soy, wheat vs bread wheat. Family, Order and specially Genera should be in ithalics (Figures 2 and 3).

Line 96: please, indicate that “SSR rRNA sequence marker” is the same than “18S rRNA sequence marker”.

Lines 108, 118 and 126: is Myke Pro Liquid or Myke Pro PS3?

Lines 110 and 128: …m2

Line 144: …(Retsch AS200 control, city, country)…

Line 145: …1.00 mm…

Line 149: …Germany…

Line 163: …(Genome Quebec Innovation Centre, city, country)…

Line 248: briefly indicate what is the meaning of higher/lower F values in LMM and LM analyses.

Line 271: please, explain what is alpha-diversity.

Line 323: …Hart et al., 2017…

Line 435: revise the reference.

Author Response

We found the comments of the the reviewer very helpful in making this major revision in which all the suggestions of the two reviewers, have been taken into account. We are very grateful to this reviewer for taking the time to think about this work seriously and provide us with the opportunity to present a more concise paper. In view of the comments, we performed new analyses, added new figures and tables, and we have modified many sections of the manuscript to answer those criticisms. This does not change the conclusions of our study but soften some of them which make the paper much stronger.

We have given the specific details below of how the manuscript has been changed or how we have addressed the reviewer’s comments.

Major points:

Some statistical analyses and data are not showed in the manuscript. Inclusion of this new data will complete the overall vision of the conclusions extracted from experiments. Specifically:

In Figure 1: include a second panel with relative abundance of taxon VTX00113.

Repeat same analysis showed in figures 2 and 3 for the different growing stages of wheat and corn crops.

Figure 5: include a second panel with PCoA analysis for growing stages of wheat and corn crops.

We added a new figures S1-4 in Supplementary materials in which we performed new analyses on relative abundance of VTX00113 and alpha-diversity. However, for simplicity, we didn't add a panel with PCoA analysis for growing stages of wheat and corn crops.

Line 242-246: provide a supplementary file with 414 unique OTUs and 68 VTs identified grouped by crops, growing stages and treatments.

We added a new Table S1 in Supplementary materials in which we provided information on 414 OTUs.

Some sections of the paper should be revised:

Lines 62-81: This paragraph fits better in the Discussion section.

We believe that this paragraph is appropriate for introduction rather than discussion.

In the Introduction section information about the microsymbiont is missing.

Added

Result section is too short and could be sorted in several subsections.

 Done

Minor points:

Unify crop names: soybean vs soy, wheat vs bread wheat. Family, Order and specially Genera should be in ithalics (Figures 2 and 3).

Done

Line 96: please, indicate that “SSR rRNA sequence marker” is the same than “18S rRNA sequence marker”.

Done

Lines 108, 118 and 126: is Myke Pro Liquid or Myke Pro PS3?

-These manes refer to different formulations of the commercial product Myke which contains panel with PCoA analysis for growing stages of wheat and corn crops

These names were correct and both products contain the same AMF isolate (R. irregulars DAMP 197198).

Lines 110 and 128: …m2

Done

Line 144: …(Retsch AS200 control, city, country)…

Done

Line 145: …1.00 mm…

Done

Line 149: …Germany…

Done

Line 163: …(Genome Quebec Innovation Centre, city, country)…

Done

Line 248: briefly indicate what is the meaning of higher/lower F values in LMM and LM analyses.

Maybe the P value is more than enough, delete the F values!

We removed F values and kept p-values

Line 271: please, explain what is alpha-diversity.

Done

Line 323: …Hart et al., 2017…

Done

Line 435: revise the reference.

Done

Round 2

Reviewer 2 Report

This paper is an amended version of the work sent previously to Microorganism. In this new manuscript the instructions, suggestions and corrections that I indicated in the first evaluation have been almost completely addressed (or at least well argued). Thus, I recommend accept the paper for publication with few modifications detailed below.

Some statistical analyses and data are not showed in the manuscript. Inclusion of this new data will complete the overall vision of the conclusions extracted from experiments. Specifically:

In Figure 1: include a second panel with relative abundance of taxon VTX00113.

Repeat same analysis showed in figures 2 and 3 for the different growing stages of wheat and corn crops.

Figure 5: include a second panel with PCoA analysis for growing stages of wheat and corn crops.

We added a new figures S1-4 in Supplementary materials in which we performed new analyses on relative abundance of VTX00113 and alpha-diversity. However, for simplicity, we didn't add a panel with PCoA analysis for growing stages of wheat and corn crops.

“Reviewer: I apologize but I meant that in Figure 1 should be include a second panel (or in a new supplementary figure) with relative abundance of taxon VTX00114, since the inoculant R. irregularis DAOM-197198 matches with both VTX00113 and VTX00114. Please, make an extra effort to include this information and sorry again for my initial mistake.

On the other hand, I think that, instead of adding a panel, you could include an additional figure with PCoA analysis for growing stages of wheat and corn crops.

Statistical tests applied in figures S1-4 should be included in their respective figure legends. Besides, a sentence highlighting what is shown in the figure should be include (such as in Fig. S2: Abundance of AMF taxa…). Figure S1: correct Inocluated in the figure and …on inoculated… in the figure legend.

Finally, information about Supplementary Materials is missing at the end of the manuscript”

Line 242-246: provide a supplementary file with 414 unique OTUs and 68 VTs identified grouped by crops, growing stages and treatments.

We added a new Table S1 in Supplementary materials in which we provided information on 414 OTUs.

“Reviewer: could you add a short explanation of information given in the first raw of the Table S1? Could you also highlight the genera of each VT in this table?

Some sections of the paper should be revised:

Lines 62-81: This paragraph fits better in the Discussion section.

We believe that this paragraph is appropriate for introduction rather than discussion.

“Reviewer: OK”

In the Introduction section information about the microsymbiont is missing.

Added

“Reviewer: OK but information about Rhizophagus irregularis DAOM-197188 is still missing”

Result section is too short and could be sorted in several subsections.

Done

“Reviewer: OK”

Unify crop names: soybean vs soy, wheat vs bread wheat. Family, Order and specially Genera should be in italics (Figures 2 and 3).

Done

“Reviewer: OK but replace soy by soybean in figures 1,2,4,5 and 6. Besides, Genera should be in italics not only in the figures but also in the text”

Line 96: please, indicate that “SSR rRNA sequence marker” is the same than “18S rRNA sequence marker”.

Done

“Reviewer: OK”

Lines 108, 118 and 126: is Myke Pro Liquid or Myke Pro PS3?

-These manes refer to different formulations of the commercial product Myke which contains panel with PCoA analysis for growing stages of wheat and corn crops

These names were correct and both products contain the same AMF isolate (R. irregulars DAMP 197198).

“Reviewer: OK”

Lines 110 and 128: …m2

Done

“Reviewer: OK”

Line 144: …(Retsch AS200 control, city, country)…

Done

“Reviewer: OK”

Line 145: …1.00 mm…

Done

“Reviewer: OK”

Line 149: …Germany…

Done

“Reviewer: OK”

Line 163: …(Genome Quebec Innovation Centre, city, country)…

Done       

“Reviewer: OK”

Line 248: briefly indicate what is the meaning of higher/lower F values in LMM and LM analyses.

Maybe the P value is more than enough, delete the F values!

We removed F values and kept p-values

“Reviewer: OK”

Line 271: please, explain what is alpha-diversity.

Done

“Reviewer: OK”

Line 323: …Hart et al., 2017…

Done

“Reviewer: OK”

Line 435: revise the reference.

Done

“Reviewer: OK”

“Additional comments:

Unify Quebéc/Québec? Elsewhere

Line 45: Reference these new sentences

Line 84: A study of the quality…

Line 108: …2) glomeromycotan diversity (alpha diversity)…

Line 124: …in a field located…

Line 126: … X-gallon tank…

Line 136:…80-gallon…

Lines 145 and 147: ..hottest and coldest days or months?

Line 178: delete Stefani et al, Unpublished?

Line 210: In how many nucleotides differ VTX00113 and VTX00114?

Line 316-317: These sentences are not well conected”

 Author Response

Our answer are in BOLD.

Some statistical analyses and data are not showed in the manuscript. Inclusion of this new data will complete the overall vision of the conclusions extracted from experiments. Specifically:

In Figure 1: include a second panel with relative abundance of taxon VTX00113.

Repeat same analysis showed in figures 2 and 3 for the different growing stages of wheat and corn crops.

Figure 5: include a second panel with PCoA analysis for growing stages of wheat and corn crops.

We added a new figures S1-4 in Supplementary materials in which we performed new analyses on relative abundance of VTX00113 and alpha-diversity. However, for simplicity, we didn't add a panel with PCoA analysis for growing stages of wheat and corn crops.

“Reviewer: I apologize but I meant that in Figure 1 should be include a second panel (or in a new supplementary figure) with relative abundance of taxon VTX00114, since the inoculant R. irregularis DAOM-197198 matches with both VTX00113 and VTX00114. Please, make an extra effort to include this information and sorry again for my initial mistake.

We added a new panel in Figure 1 with relative abundance of taxon VTX00114.

On the other hand, I think that, instead of adding a panel, you could include an additional figure with PCoA analysis for growing stages of wheat and corn crops.

We believe the adding a panel in figure 1 is easier for readers (see the revised figure 1).

Statistical tests applied in figures S1-4 should be included in their respective figure legends. Besides, a sentence highlighting what is shown in the figure should be include (such as in Fig. S2: Abundance of AMF taxa…). Figure S1: correct Inocluated in the figure and …on inoculated… in the figure legend.

We have corrected and added information in all supple figures as suggested by the reviewer.

Finally, information about Supplementary Materials is missing at the end of the manuscript”

Added

Line 242-246: provide a supplementary file with 414 unique OTUs and 68 VTs identified grouped by crops, growing stages and treatments.

We added a new Table S1 in Supplementary materials in which we provided information on 414 OTUs.

“Reviewer: could you add a short explanation of information given in the first raw of the Table S1? Could you also highlight the genera of each VT in this table?

 We added description of Table S1. However, we couldn’t highlight the genera of each VT in this table.

Some sections of the paper should be revised:

Lines 62-81: This paragraph fits better in the Discussion section.

We believe that this paragraph is appropriate for introduction rather than discussion.

“Reviewer: OK”

In the Introduction section information about the microsymbiont is missing.

Added

“Reviewer: OK but information about Rhizophagus irregularis DAOM-197188 is still missing”

 We added a sentence on Rhizophagus irregularis DAOM-197198.

Result section is too short and could be sorted in several subsections.

Done

“Reviewer: OK”

Unify crop names: soybean vs soy, wheat vs bread wheat. Family, Order and specially Genera should be in italics (Figures 2 and 3).

Done

“Reviewer: OK but replace soy by soybean in figures 1,2,4,5 and 6. Besides, Genera should be in italics not only in the figures but also in the text”

 All figures were changed by replacing soy by soybean. Genera ganged in italics in the text.

Line 96: please, indicate that “SSR rRNA sequence marker” is the same than “18S rRNA sequence marker”.

Done

“Reviewer: OK”

Lines 108, 118 and 126: is Myke Pro Liquid or Myke Pro PS3?

-These manes refer to different formulations of the commercial product Myke which contains panel with PCoA analysis for growing stages of wheat and corn crops

These names were correct and both products contain the same AMF isolate (R. irregulars DAMP 197198).

“Reviewer: OK”

Lines 110 and 128: …m2

Done

“Reviewer: OK”

Line 144: …(Retsch AS200 control, city, country)…

Done

“Reviewer: OK”

Line 145: …1.00 mm…

Done

“Reviewer: OK”

Line 149: …Germany…

Done

“Reviewer: OK”

Line 163: …(Genome Quebec Innovation Centre, city, country)…

Done       

“Reviewer: OK”

Line 248: briefly indicate what is the meaning of higher/lower F values in LMM and LM analyses.

We removed F values and kept p-values

“Reviewer: OK”

Line 271: please, explain what is alpha-diversity.

Done

“Reviewer: OK”

Line 323: …Hart et al., 2017…

Done

“Reviewer: OK”

Line 435: revise the reference.

Done

“Reviewer: OK”

“Additional comments:

Unify Quebéc/Québec? Elsewhere

Done, we used Québec throughout the text.

Line 45: Reference these new sentences

The following reference was added: Bonfante, P.; Genre, A. Plants and arbuscular mycorrhizal fungi: an evolutionary-developmental perspective. Trends in plant science 2008, 13, 492–498

Line 84: A study of the quality…

Corrected.

Line 108: …2) glomeromycotan diversity (alpha diversity)…

Corrected.

Line 124: …in a field located…

Corrected.

Line 126: … X-gallon tank…

Corrected.

Line 136:…80-gallon…

Lines 145 and 147: ..hottest and coldest days or months?

Days

Line 178: delete Stefani et al, Unpublished?

Deleted

Line 210: In how many nucleotides differ VTX00113 and VTX00114?

One nucleotide difference (Lines 373-374).

Line 316-317: These sentences are not well conected”

We rephrased the sentence.